# Robust learning of low-dimensional dynamics from large neural ensembles

David Pfau        Eftychios A. Pnevmatikakis        Liam Paninski

Center for Theoretical Neuroscience
Department of Statistics
Grossman Center for the Statistics of Mind
Columbia University, New York, NY
`pfau@neurotheory.columbia.edu`
`{eftychios,liam}@stat.columbia.edu`

## Abstract

Recordings from large populations of neurons make it possible to search for hypothesized low-dimensional dynamics. Finding these dynamics requires models that take into account biophysical constraints and can be fit efficiently and robustly. Here, we present an approach to dimensionality reduction for neural data that is convex, does not make strong assumptions about dynamics, does not require averaging over many trials and is extensible to more complex statistical models that combine local and global influences. The results can be combined with spectral methods to learn dynamical systems models. The basic method extends PCA to the exponential family using nuclear norm minimization. We evaluate the effectiveness of this method using an exact decomposition of the Bregman divergence that is analogous to variance explained for PCA. We show on model data that the parameters of latent linear dynamical systems can be recovered, and that even if the dynamics are not stationary we can still recover the true latent subspace. We also demonstrate an extension of nuclear norm minimization that can separate sparse local connections from global latent dynamics. Finally, we demonstrate improved prediction on real neural data from monkey motor cortex compared to fitting linear dynamical models without nuclear norm smoothing.

## 1 Introduction

Progress in neural recording technology has made it possible to record spikes from ever larger populations of neurons [1]. Analysis of these large populations suggests that much of the activity can be explained by simple population-level dynamics [2]. Typically, this low-dimensional activity is extracted by principal component analysis (PCA) [3, 4, 5], but in recent years a number of extensions have been introduced in the neuroscience literature, including jPCA [6] and demixed principal component analysis (dPCA) [7]. A downside of these methods is that they do not treat either the discrete nature of spike data or the positivity of firing rates in a statistically principled way. Standard practice smooths the data substantially or averages it over many trials, losing information about fine temporal structure and inter-trial variability.

One alternative is to fit a more complex statistical model directly from spike data, where temporal dependencies are attributed to latent low dimensional dynamics [8, 9]. Such models can account for the discreteness of spikes by using point-process models for the observations, and can incorporate temporal dependencies into the latent state model. State space models can include complex interactions such as switching linear dynamics [10] and direct coupling between neurons [11]. These methods have drawbacks too: they are typically fit by approximate EM [12] or other methods that are prone to local minima, the number of latent dimensions is typically chosen ahead of time, and a certain class of possible dynamics must be chosen before doing dimensionality reduction.

In this paper we attempt to combine the computational tractability of PCA and related methods with the statistical richness of state space models. Our approach is convex and based on recent advances in system identification using *nuclear norm minimization* [13, 14, 15], a convex relaxation of matrix rank minimization. Compared to recent work on spectral methods for fitting state space models [16], our method more easily generalizes to handle different nonlinearities, non-Gaussian, non-linear, and non-stationary latent dynamics, and direct connections between observed neurons. When applied to model data, we find that: (1) low-dimensional subspaces can be accurately recovered, even when the dynamics are unknown and nonstationary (2) standard spectral methods can robustly recover the parameters of state space models when applied to data projected into the recovered subspace (3) the confounding effects of common input for inferring sparse synaptic connectivity can be ameliorated by accounting for low-dimensional dynamics. In applications to real data we find comparable performance to models trained by EM with less computational overhead, particularly as the number of latent dimensions grows.

The paper is organized as follows. In section 2 we introduce the class of models we aim to fit, which we call low-dimensional generalized linear models (LD-GLM). In section 3 we present a convex formulation of the parameter learning problem for these models, as well as a generalization of variance explained to LD-GLMs used for evaluating results. In section 4 we show how to fit these models using the *alternating direction method of multipliers* (ADMM). In section 5 we present results on real and artificial neural datasets. We discuss the results and future directions in section 6.

## 2   Low dimensional generalized linear models

Our model is closely related to the generalized linear model (GLM) framework for neural data [17]. Unlike the standard GLM, where the inputs driving the neurons are observed, we assume that the driving activity is unobserved, but lies on some low dimensional subspace. This can be a useful way of capturing spontaneous activity, or accounting for strong correlations in large populations of neurons. Thus, instead of fitting a linear receptive field, the goal of learning in low-dimensional GLMs is to accurately recover the latent subspace of activity.

Let $x_t \in \mathbb{R}^m$ be the value of the dynamics at time $t$. To turn this into spiking activity, we project this into the space of neurons: $y_t = Cx_t + b$ is a vector in $\mathbb{R}^n$, $n \gg m$, where each dimension of $y_t$ corresponds to one neuron. $C \in \mathbb{R}^{n \times m}$ denotes the subspace of the neural population and $b \in \mathbb{R}^n$ the bias vector for all the neurons. As $y_t$ can take on negative values, we cannot use this directly as a firing rate, and so we pass each element of $y_t$ through some convex and log-concave increasing point-wise nonlinearity $f : \mathbb{R} \to \mathbb{R}_+$. Popular choices for nonlinearities include $f(x) = \exp(x)$ and $f(x) = \log(1 + \exp(x))$. To account for biophysical effects such as refractory periods, bursting, and direct synaptic connections, we include a linear dependence on spike history before the nonlinearity. The firing rate $f(y_t)$ is used as the rate for some point process $\xi$ such as a Poisson process to generate a vector of spike counts $s_t$ for all neurons at that time:

$$y_t \;\; = \;\; Cx_t + \sum_{\tau=1}^{k} D_\tau s_{t-\tau} + b \tag{1}$$

$$s_t \;\; \sim \;\; \xi(f(y_t)) \tag{2}$$

Much of this paper is focused on estimating $y_t$, which is the natural parameter for the Poisson distribution in the case $f(\cdot) = \exp(\cdot)$, and so we refer to $y_t$ as the *natural* rate to avoid confusion with the *actual* rate $f(y_t)$. We will see that our approach works with any point process with a log-concave likelihood, not only Poisson processes.

We can extend this simple model by adding dynamics to the low-dimensional latent state, including input-driven dynamics. In this case the model is closely related to the common input model used in neuroscience [11], the difference being that the observed input is added to $x_t$ rather than being directly mapped to $y_t$. The case without history terms and with linear Gaussian dynamics is a well-studied state space model for neural data, usually fit by EM [19, 12, 20], though a consistent spectral method has been derived [16] for the case $f(\cdot) = \exp(\cdot)$. Unlike these methods, our approach largely decouples the problem of dimensionality reduction and learning dynamics: even in the case of nonstationary, non-Gaussian dynamics where $A$, $B$ and Cov$[\epsilon]$ change over time, we can still robustly recover the latent subspace spanned by $x_t$.

# 3 Learning

## 3.1 Nuclear norm minimization

In the case that the spike history terms $D_{1:k}$ are zero, the natural rate at time $t$ is $y_t = Cx_t + b$, so all $y_t$ are elements of some $m$-dimensional affine space given by the span of the columns of $C$ offset by $b$. Ideally, our estimate of $y_{1:T}$ would trade off between making the dimension of this affine space as low as possible and the likelihood of $y_{1:T}$ as high as possible. Let $Y = [y_1, \ldots, y_T]$ be the $n \times T$ matrix of natural rates and let $\mathcal{A}(\cdot)$ be the row mean centering operator $\mathcal{A}(Y) = Y - \frac{1}{T}Y\mathbf{1}_T\mathbf{1}_T^T$. Then $\text{rank}(\mathcal{A}(Y)) = m$. Ideally we would minimize $\lambda nT\text{rank}(\mathcal{A}(Y)) - \sum_{t=1}^T \log p(s_t|y_t)$, where $\lambda$ controls how much we trade off between a simple solution and the likelihood of the data, however general rank minimization is a hard non convex problem. Instead we replace the matrix rank with its convex envelope: the sum of singular values or *nuclear norm* $\|\cdot\|_*$ [13], which can be seen as the analogue of the $\ell_1$ norm for vector sparsity. Our problem then becomes:

$$\min_Y \lambda\sqrt{nT}\|\mathcal{A}(Y)\|_* - \sum_{t=1}^T \log p(s_t|y_t) \tag{3}$$

Since the log likelihood scales linearly with the size of the data, and the singular values scale with the square root of the size, we also add a factor of $\sqrt{nT}$ in front of the nuclear norm term. In the examples in this paper, we assume spikes are drawn from a Poisson distribution:

$$\log p(s_t|y_t) = \sum_{i=1}^N s_{it}\log f(y_{it}) - f(y_{it}) - \log s_{it}! \tag{4}$$

However, this method can be used with any point process with a log-concave likelihood. This can be viewed as a convex formulation of exponential family PCA [21, 22] which does not fix the number of principal components ahead of time.

## 3.2 Stable principal component pursuit

The model above is appropriate for cases where the spike history terms $D_\tau$ are zero, that is the observed data can entirely be described by some low-dimensional global dynamics. In real data neurons exhibit history-dependent behavior like bursting and refractory periods. Moreover if the recorded neurons are close to each other some may have direct synaptic connections. In this case $D_\tau$ may have full column rank, so from Eq. 1 it is clear that $y_t$ is no longer restricted to a low-dimensional affine space. In most practical cases we expect $D_\tau$ to be sparse, since most neurons are not connected to one another. In this case the natural rates matrix combines a low-rank term and a sparse term, and we can minimize a convex function that trades off between the rank of one term via the nuclear norm, the sparsity of another via the $\ell_1$ norm, and the data log likelihood:

$$\min_{Y,D_{1:k},L} \lambda\sqrt{nT}\|\mathcal{A}(L)\|_* + \gamma\frac{T}{n}\sum_{\tau=1}^k \|D_\tau\|_1 - \sum_{t=1}^T \log p(s_t|y_t) \tag{5}$$

$$\text{s.t. } Y = L + \sum_{\tau=1}^k D_\tau S_\tau, \text{ with } S_\tau = [0_{n,\tau}, s_1, \ldots, s_{T-\tau}],$$

where $0_{n,\tau}$ is a matrix of zeros of size $n \times \tau$, used to account for boundary effects. This is an extension of *stable principal component pursuit* [23], which separates sparse and low-rank components of a noise-corrupted matrix. Again to ensure that every term in the objective function of Eq. 5 has roughly the same scaling $\mathcal{O}(nT)$ we have multiplied each $\ell_1$ norm with $T/n$. One can also consider the use of a group sparsity penalty where each group collects a specific synaptic weight across all the $k$ time lags.

## 3.3 Evaluation through Bregman divergence decomposition

We need a way to evaluate the model on held out data, without assuming a particular form for the dynamics. As we recover a subspace spanned by the columns of $Y$ rather than a single parameter, this presents a challenge. One option is to compute the marginal likelihood of the data integrated

over the entire subspace, but this is computationally difficult. For the case of PCA, we can project the held out data onto a subspace spanned by principal components and compute what fraction of total variance is explained by this subspace. We extend this approach beyond the linear Gaussian case by use of a generalized Pythagorean theorem.

For any exponential family with natural parameters $\theta$, link function $g$, function $F$ such that $\nabla F = g^{-1}$ and sufficient statistic $T$, the log likelihood can be written as $D_F[\theta||g(T(x))] - h(x)$, where $D.[\cdot||\cdot]$ is a Bregman divergence [24]: $D_F[x||y] = F(x) - F(y) - (x - y)^T \nabla F(y)$. Intuitively, the Bregman divergence between $x$ and $y$ is the difference between the value of $F(x)$ and the value of the best linear approximation around $y$. Bregman divergences obey a generalization of the Pythagorean theorem: for any affine set $\Omega$ and points $x \notin \Omega$ and $y \in \Omega$, it follows that $D_F[x||y] = D_F[x||\Pi_\Omega(x)] + D_F[\Pi_\Omega(x)||y]$ where $\Pi_\Omega(x) = \arg\min_{\omega \in \Omega} D_F[x||\omega]$ is the projection of $x$ onto $\Omega$. In the case of squared error this is just a linear projection, and for the case of GLM log likelihoods this is equivalent to maximum likelihood estimation when the natural parameters are restricted to $\Omega$.

Given a matrix of natural rates recovered from training data, we compute the *fraction of Bregman divergence explained* by a sequence of subspaces as follows. Let $u_i$ be the $i$th singular vector of the recovered natural rates. Let $b$ be the mean natural rate, and let $y_t^{(q)}$ be the maximum likelihood natural rates restricted to the space spanned by $u_1, \ldots, u_q$:

$$y_t^{(q)} = \sum_{i=1}^{q} u_i v_{it}^{(q)} + \sum_{\tau=1}^{k} D_\tau s_{t-\tau} + b$$

$$v_t^{(q)} = \arg\max_v \log p\left(s_t \middle| \sum_{i=1}^{q} u_i v_{it} + \sum_{\tau=1}^{k} D_\tau s_{t-\tau} + b\right) \tag{6}$$

Here $v_t^{(q)}$ is the projection of $y_t^{(q)}$ onto the singular vectors. Then the divergence from the mean explained by the $q$th dimension is given by

$$\frac{\sum_t D_F\left[y_t^{(q-1)} \middle|\middle| y_t^{(q)}\right]}{\sum_t D_F\left[y_t^{(0)} \middle|\middle| g(s_t)\right]} \tag{7}$$

where $y_t^{(0)}$ is the bias $b$ plus the spike history terms. The sum of divergences explained over all $q$ is equal to one by virtue of the generalized Pythagorean theorem. For Gaussian noise $g(x) = x$ and $F(x) = \frac{1}{2}||x||^2$ and this is exactly the variance explained by each principal component, while for Poisson noise $g(x) = \log(x)$ and $F(x) = \sum_i \exp(x_i)$. This decomposition is only exact if $f = g^{-1}$ in Eq. 4, that is, if the nonlinearity is exponential. However, for other nonlinearities this may still be a useful approximation, and gives us a principled way of evaluating the goodness of fit of a learned subspace.

## 4 Algorithms

Minimizing Eq. 3 and Eq. 5 is difficult, because the nuclear and $\ell_1$ norm are not differentiable everywhere. By using the *alternating direction method of multipliers* (ADMM), we can turn these problems into a sequence of tractable subproblems [25]. While not always the fastest method for solving a particular problem, we use it for its simplicity and generality. We describe the algorithm below, with more details in the supplemental materials.

### 4.1 Nuclear norm minimization

To find the optimal $Y$ we alternate between minimizing an augmented Lagrangian with respect to $Y$, minimizing with respect to an auxiliary variable $Z$, and performing gradient ascent on a Lagrange multiplier $\Lambda$. The augmented Lagrangian is

$$\mathcal{L}_\rho(Y, Z, \Lambda) = \lambda\sqrt{nT}||Z||_* - \sum_t \log p(s_t|y_t) + \langle \Lambda, \mathcal{A}(Y) - Z \rangle + \frac{\rho}{2}||\mathcal{A}(Y) - Z||_F^2 \tag{8}$$

which is a smooth function of $Y$ and can be minimized by Newton's method. The gradient and Hessian of $\mathcal{L}_\rho$ with respect to $Y$ at iteration $k$ are

$$\nabla_Y \mathcal{L}_\rho = -\nabla_Y \sum_t \log p(s_t|y_t) + \rho \mathcal{A}(Y) - \mathcal{A}^T(\rho Z_k - \Lambda_k) \tag{9}$$

$$\nabla_Y^2 \mathcal{L}_\rho = -\nabla_Y^2 \sum_t \log p(s_t|y_t) + \rho I_{nT} - \rho \frac{1}{T}(\mathbf{1}_T \otimes I_n)(\mathbf{1}_T \otimes I_n)^T \tag{10}$$

where $\otimes$ is the Kronecker product. Note that the first two terms of the Hessian are diagonal and the third is low-rank, so the Newton step can be computed in $\mathcal{O}(nT)$ time by using the Woodbury matrix inversion lemma.

The minimum of Eq. 17 with respect to $Z$ is given exactly by singular value thresholding:

$$Z_{k+1} = U \mathcal{S}_{\lambda\sqrt{nT}/\rho}(\Sigma)V^T, \tag{11}$$

where $U\Sigma V^T$ is the singular value decomposition of $\mathcal{A}(Y_{k+1}) + \Lambda_k/\rho$, and $\mathcal{S}_t(\cdot)$ is the (pointwise) soft thresholding operator $\mathcal{S}_t(x) = \mathrm{sgn}(x)\max(0, |x| - t)$. Finally, the update to $\Lambda$ is a simple gradient ascent step: $\Lambda_{k+1} = \Lambda_k + \rho(\mathcal{A}(Y_{k+1}) - Z_{k+1})$ where $\rho$ is a step size that can be chosen.

## 4.2   Stable principal component pursuit

To extend ADMM to the problem in Eq. 5 we only need to add one extra step, taking the minimum over the connectivity matrices with the other parameters held fixed. To simplify the notation, we group the connectivity matrices into a single matrix $D = (D_1, \ldots, D_k)$, and stack the different time-shifted matrices of spike histories on top of one another to form a single spike history matrix $H$. The objective then becomes

$$\min_{Y,D} \lambda\sqrt{nT}||\mathcal{A}(Y - DH)||_* + \gamma\frac{T}{n}||D||_1 - \sum_t \log p(s_t|y_t) \tag{12}$$

where we have substituted $Y - DH$ for the variable $L$, and the augmented Lagrangian is

$$\mathcal{L}_\rho(Y, Z, D, \Lambda) = \lambda\sqrt{nT}||Z||_* + \gamma\frac{T}{n}||D||_1 - \sum_t \log p(s_t|y_t) \tag{13}$$
$$+ \langle \Lambda, \mathcal{A}(Y - DH) - Z \rangle + \frac{\rho}{2}||\mathcal{A}(Y - DH) - Z||_F^2$$

The updates for $\Lambda$ and $Z$ are almost unchanged, except that $\mathcal{A}(Y)$ becomes $\mathcal{A}(Y - DH)$. Likewise for $Y$ the only change is one additional term in the gradient:

$$\nabla_Y \mathcal{L}_\rho = -\nabla_Y \sum_t \log p(s_t|y_t) + \rho \mathcal{A}(Y) - \mathcal{A}^T(\rho Z + \rho \mathcal{A}(DH) - \Lambda) \tag{14}$$

Minimizing $D$ requires solving:

$$\arg\min_D \gamma\frac{T}{n}||D||_1 + \frac{\rho}{2}||\mathcal{A}(DH) + Z - \mathcal{A}(Y) - \Lambda/\rho||_F^2 \tag{15}$$

This objective has the same form as LASSO regression. We solve this using ADMM as well, but any method for LASSO regression can be substituted.

# 5   Experiments

We demonstrate our method on a number of artificial datasets and one real dataset. First, we show in the absence of spike history terms that the true low dimensional subspace can be recovered in the limit of large data, even when the dynamics are nonstationary. Second, we show that spectral methods can accurately recover the transition matrix when dynamics are linear. Third, we show that local connectivity can be separated from low-dimensional common input. Lastly, we show that nuclear-norm penalized subspace recovery leads to improved prediction on real neural data recorded from macaque motor cortex.

Model data was generated with 8 latent dimension and 200 neurons, without any external input. For linear dynamical systems, the transition matrix was sampled from a Gaussian distribution, and the

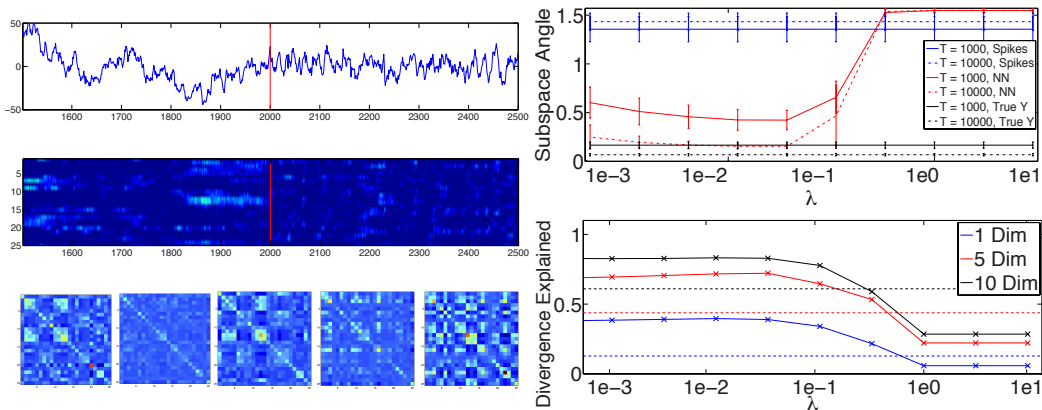

Figure 1: Recovering low-dimensional subspaces from nonstationary model data. While the subspace remains the same, the dynamics switch between 5 different linear systems. Left top: one dimension of the latent trajectory, switching from one set of dynamics to another (red line). Left middle: firing rates of a subset of neurons during the same switch. Left bottom: covariance between spike counts for different neurons during each epoch of linear dynamics. Right top: Angle between the true subspace and top principal components directly from spike data, from natural rates recovered by nuclear norm minimization, and from the true natural rates. Right bottom: fraction of Bregman divergence explained by the top 1, 5 or 10 dimensions from nuclear norm minimization. Dotted lines are variance explained by the same number of principal components. For $\lambda < 0.1$ the divergence explained by a given number of dimensions exceeds the variance explained by the same number of PCs.

eigenvalues rescaled so the magnitude fell between .9 and .99 and the angle between $\pm\frac{\pi}{10}$, yielding slow and stable dynamics. The linear projection $C$ was a random Gaussian matrix with standard deviation 1/3, and the biases $b_i$ were sampled from $\mathcal{N}(-4, 1)$, which we found gave reasonable firing rates with nonlinearity $f(x) = \log(1 + \exp(x))$. To investigate the variance of our estimates, we generated multiple trials of data with the same parameters but different innovations.

We first sought to show that we could accurately recover the subspace in which the dynamics take place even when those dynamics are not stationary. We split each trial into 5 epochs and in each epoch resampled the transition matrix $A$ and set the covariance of innovations $\epsilon_t$ to $QQ^T$ where $Q$ is a random Gaussian matrix. We performed nuclear norm minimization on data generated from this model, varying the smoothing parameter $\lambda$ from $10^{-3}$ to 10, and compared the subspace angle between the top 8 principal components and the true matrix $C$. We repeated this over 10 trials to compute the variance of our estimator. We found that when smoothing was optimized the recovered subspace was significantly closer to the true subspace than the top principal components taken directly from spike data. Increasing the amount of data from 1000 to 10000 time bins significantly reduced the average subspace angle at the optimal $\lambda$. The top PCs of the true natural rates $Y$, while not spanning exactly the same space as $C$ due to differences between the mean column and true bias $b$, was still closer to the true subspace than the result of nuclear norm minimization.

We also computed the fraction of Bregman divergence explained by the sequence of spaces spanned by successive principal components, solving Eq. 6 by Newton's method. We did not find a clear drop at the true dimensionality of the subspace, but we did find that a larger share of the divergence could be explained by the top dimensions than by PCA directly on spikes. Results are presented in Fig. 1.

To show that the parameters of a latent dynamical system can be recovered, we investigated the performance of spectral methods on model data with linear Gaussian latent dynamics. As the model is a linear dynamical system with GLM output, we call this a GLM-LDS model. After estimating natural rates by nuclear norm minimization with $\lambda = 0.01$ on 10 trials of 10000 time bins with unit-variance innovations $\epsilon_t$, we fit the transition matrix $A$ by subspace identification (SSID) [26]. The transition matrix is only identifiable up to a change of coordinates, so we evaluated our fit by comparing the eigenvalues of the true and estimated $A$. Results are presented in Fig. 2. As expected, SSID directly on spikes led to biased estimates of the transition. By contrast, SSID on the output of

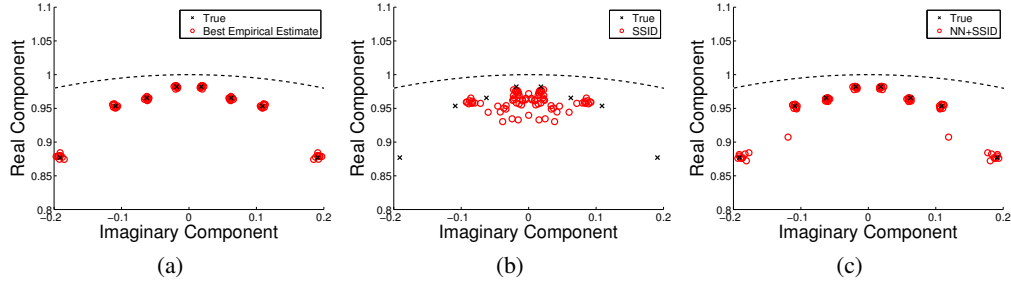

(a)                    (b)                    (c)

Figure 2: Recovered eigenvalues for the transition matrix of a linear dynamical system from model neural data. Black: true eigenvalues. Red: recovered eigenvalues. (2a) Eigenvalues recovered from the true natural rates. (2b) Eigenvalues recovered from subspace identification directly on spike counts. (2c) Eigenvalues recovered from subspace identification on the natural rates estimated by nuclear norm minimization.

nuclear norm minimization had little bias, and seemed to perform almost as well as SSID directly on the true natural rates. We found that other methods for fitting linear dynamical systems from the estimated natural rates were biased, as was SSID on the result of nuclear norm minimization *without* mean-centering (see the supplementary material for more details).

We incorporated spike history terms into our model data to see whether local connectivity and global dynamics could be separated. Our model network consisted of 50 neurons, randomly connected with 95% sparsity, and synaptic weights sampled from a unit variance Gaussian. Data were sampled from 10000 time bins. The parameters $\lambda$ and $\gamma$ were both varied from $10^{-10}$ to $10^4$. We found that we could recover synaptic weights with an $r^2$ up to .4 on this data by combining both a nuclear norm and $\ell_1$ penalty, compared to at most .25 for an $\ell_1$ penalty alone, or 0.33 for a nuclear norm penalty alone. Somewhat surprisingly, at the extreme of either no nuclear norm penalty or a dominant nuclear norm penalty, increasing the $\ell_1$ penalty never improved estimation. This suggests that in a regime with strong common inputs, some kind of correction is necessary not only for sparse penalties to achieve optimal performance, but to achieve any improvement over maximum likelihood. It is also of interest that the peak in $r^2$ is near a sharp transition to total sparsity.

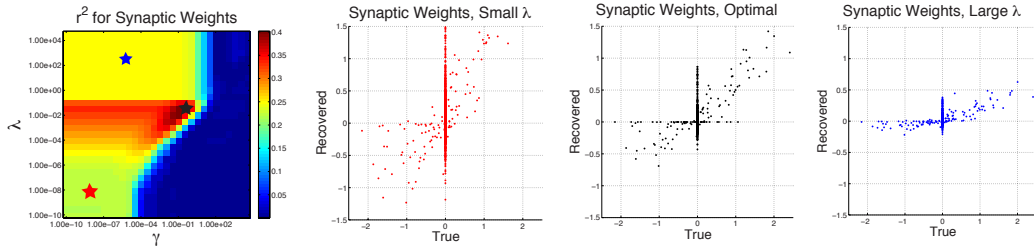

Figure 3: Connectivity matrices recovered by SPCP on model data. Left: $r^2$ between true and recovered synaptic weights across a range of parameters. The position in parameter space of the data to the right is highlighted by the stars. Axes are on a log scale. Right: scatter plot of true versus recovered synaptic weights, illustrating the effect of the nuclear norm term.

Finally, we demonstrated the utility of our method on real recordings from a large population of neurons. The data consists of 125 well-isolated units from a multi-electrode recording in macaque motor cortex while the animal was performing a pinball task in two dimensions. Previous studies on this data [27] have shown that information about arm velocity can be reliably decoded. As the electrodes are spaced far apart, we do not expect any direct connections between the units, and so leave out the $\ell_1$ penalty term from the objective. We used 800 seconds of data binned every 100 ms for training and 200 seconds for testing. We fit linear dynamical systems by subspace identification as in Fig. 2, but as we did not have access to a "true" linear dynamical system for comparison, we evaluated our model fits by approximating the held out log likelihood by Laplace-Gaussian filtering [28].

We also fit the GLM-LDS model by running randomly initialized EM for 50 iterations for models with up to 30 latent dimensions (beyond which training was prohibitively slow). We found that a strong nuclear norm penalty improved prediction by several hundred bits per second, and that fewer dimensions were needed for optimal prediction as the nuclear norm penalty was increased. The best fit models predicted held out data nearly as well as models trained via EM, even though nuclear norm minimization is not directly maximizing the likelihood of a linear dynamical system.

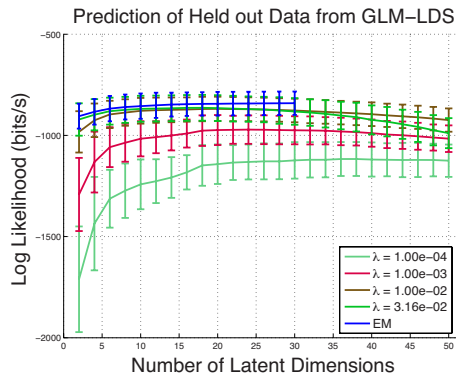

Figure 4: Log likelihood of held out motor cortex data versus number of latent dimensions for different latent linear dynamical systems. Prediction improves as $\lambda$ increases, until it is comparable to EM.

## 6 Discussion

The method presented here has a number of straightforward extensions. If the dimensionality of the latent state is *greater* than the dimensionality of the data, for instance when there are long-range history dependencies in a small population of neurons, we would extend the natural rate matrix $Y$ so that each column contains multiple time steps of data. $Y$ is then a *block-Hankel matrix*. Constructing the block-Hankel matrix is also a linear operation, so the objective is still convex and can be efficiently minimized [15]. If there are also observed inputs $u_t$ then the term inside the nuclear norm should also include a projection orthogonal to the row space of the inputs. This could enable joint learning of dynamics and receptive fields for small populations of neurons with high dimensional inputs.

Our model data results on connectivity inference have important implications for practitioners working with highly correlated data. GLM models with sparsity penalties have been used to infer connectivity in real neural networks [29], and in most cases these networks are only partially observed and have large amounts of common input. We offer one promising route to removing the confounding influence of unobserved correlated inputs, which explicitly models the common input rather than conditioning on it [30].

It remains an open question what kinds of dynamics can be learned from the recovered natural parameters. In this paper we have focused on linear systems, but nuclear norm minimization could just as easily be combined with spectral methods for switching linear systems and general nonlinear systems. We believe that the techniques presented here offer a powerful, extensible and robust framework for extracting structure from neural activity.

### Acknowledgments

Thanks to Zhang Liu, Michael C. Grant, Lars Buesing and Maneesh Sahani for helpful discussions, and Nicho Hatsopoulos for providing data. This research was generously supported by an NSF CAREER grant.

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
