[Supplementary Material · pfau-nips2013-supplement.pdf]

# A   Alternating Direction Method of Multipliers

We present here a brief overview of the alternating direction method of multipliers (ADMM), along with a derivation of the algorithms presented in the text and details on the convergence criteria that were omitted from the main text. Here $X^i$ denotes the $i$th row of the matrix $X$.

ADMM is a method for solving problems of the form:

$$\min_Y f(Y) + g(Y) \tag{16}$$

where $f(\cdot)$ and $g(\cdot)$ are both convex, but not necessarily differentiable everywhere. We introduce an auxiliary variable $Z$, a Lagrange multiplier $\Lambda$, and an augmented term that depends on a learning rate $\rho$ to form the augmented Lagrangian:

$$\mathcal{L}_\rho(Y, Z, \Lambda) \triangleq f(Y) + g(Z) + \langle \Lambda, Y - Z \rangle + \frac{\rho}{2}||Y - Z||_F^2 \tag{17}$$

If we minimize $\mathcal{L}_\rho$ with respect to $Y$ and $Z$ the result is a concave function of $\Lambda$ (the convex conjugate or Legendre-Fenchel transform), and the value of $Y$ at the solution to $\max_\Lambda \inf_{Y,Z} \mathcal{L}_\rho$ is also the solution to Eq. 16. At this solution the augmented term $\frac{\rho}{2}||Y - Z||_F^2$ vanishes; it is there to guarantee that $\inf_{Y,Z} \mathcal{L}_\rho$ is well behaved before convergence.

ADMM does not directly maximize $\inf_{Y,Z} \mathcal{L}_\rho$. Instead, it alternates between minimizing $Y$, minimizing $Z$, and gradient ascent on $\Lambda$:

$$Y_{k+1} = \arg\min_Y \mathcal{L}_\rho(Y, Z_k, \Lambda_k) \tag{18}$$

$$Z_{k+1} = \arg\min_Z \mathcal{L}_\rho(Y_{k+1}, Z, \Lambda_k) \tag{19}$$

$$\Lambda_{k+1} = \Lambda_k + \rho(Y_{k+1} - Z_{k+1}) \tag{20}$$

This is guaranteed to converge to the global solution of Eq. 16.

## A.1   Derivation and Algorithm for Nuclear Norm Minimization

When applied to Eq. 3, the augmented Lagrangian takes the form:

$$\mathcal{L}_\rho(Y, Z, \Lambda) = \lambda\sqrt{nT}||Z||_* - \sum_t \log p(s_t|y_t) + \langle \Lambda, \mathcal{A}(Y) - Z \rangle + \frac{\rho}{2}||\mathcal{A}(Y) - Z||_F^2 \tag{21}$$

Note that our equality constraint is slightly different: $\mathcal{A}(Y) = Z$ instead of $Y = Z$. It turns out that in this form, Eq. 19 has an exact solution. From Eq. 20, the update to $\Lambda$ is clearly the simplest: $\Lambda_{k+1} = \Lambda_k + \rho(\mathcal{A}(Y_{k+1}) - Z_{k+1})$. The Hessian of Eq. 21 with respect to $Y$ is given by

$$\nabla_Y^2 \mathcal{L}_\rho = -\nabla_Y^2 \sum_t \log p(s_t|y_t) + \rho\mathcal{A}^T\mathcal{A} \tag{22}$$

where $\mathcal{A}^T(\cdot)$ is the transpose of the operator $\mathcal{A}(\cdot)$ and $\mathcal{A}^T\mathcal{A}$ is the product of the operator and its transpose written in matrix form. The Newton search direction $-(\nabla_Y^2\mathcal{L}_\rho)^{-1}\nabla_Y\mathcal{L}_\rho$ can be computed efficiently by exploiting the structure of the Hessian. The Hessian of the log likelihood term is diagonal, since the likelihood of the data $s_{it}$ for neuron $i$ at time $t$ only depends on $y_{it}$. Moreover, if $\text{vec}(\cdot)$ denotes the vectorizing operator then the mean centering operator $\mathcal{A}(\cdot)$ can be expressed as

$$\text{vec}(\mathcal{A}(Y)) = \left( I_{nT} - \frac{1}{T}(\mathbf{1}_T \otimes I_n)(\mathbf{1}_T \otimes I_n)^T \right) \text{vec}(Y), \tag{23}$$

It follows that $\mathcal{A}$ is self-adjoint and idempotent, and the Hessian simplifies to

$$\nabla_Y^2 \mathcal{L}_\rho = -\nabla_Y^2 \sum_t \log p(s_t|y_t) + \rho I_{nT} - \rho\frac{1}{T}(\mathbf{1}_T \otimes I_n)(\mathbf{1}_T \otimes I_n)^T \tag{24}$$

which is the sum of a diagonal term, $-\nabla_Y^2 \sum_t \log p(s_t|y_t) + \rho I_{nT}$, and the term, $-\rho\frac{1}{T}(\mathbf{1}_T \otimes I_n)(\mathbf{1}_T \otimes I_n)^T$, which is only rank $n$ rather than $nT$. Let $D$ be the diagonal part of the Hessian

$$D = -\nabla_Y^2 \sum_t \log p(s_t|y_t) + \rho I_{nT}. \tag{25}$$

Using the Woodbury lemma we have

$$(\nabla_Y^2 \mathcal{L}_\rho)^{-1} = D^{-1} + D^{-1}(\mathbf{1}_T \otimes I_n)((T/\rho)I_n - (\mathbf{1}_T \otimes I_n)^T D^{-1}(\mathbf{1}_T \otimes I_n))^{-1}(\mathbf{1}_T \otimes I_n)^T D^{-1}. \quad (26)$$

Now let $\boldsymbol{d} = \mathrm{diag}\{D^{-1}\}$ and $\Delta$ the $n \times T$ matrix such that $\mathrm{vec}(\Delta) = \boldsymbol{d}$. A quick calculation shows that the matrix $(\mathbf{1}_T \otimes I_n)^T D^{-1}(\mathbf{1}_T \otimes I_n)$ is diagonal, with its diagonal equal to $\Delta \mathbf{1}_T$. It follows that the Newton direction $-(\nabla_Y^2 \mathcal{L}_\rho)^{-1}\nabla_Y \mathcal{L}_\rho$ can be computed efficiently in $\mathcal{O}(nT)$ time and with $\mathcal{O}(nT)$ memory requirements, without having to explicitly construct the Hessian.

---

**Algorithm 1** Alternating Direction Method of Multipliers for Nuclear Norm minimization without connectivity (Eq. 3)

---

   **input** Matrix of spike counts $S$, learning rate $\rho$, parameter $\lambda$
   $Y \leftarrow \log(S+1), Z \leftarrow 0, \Lambda \leftarrow 0$
   **while** $r_p > \epsilon_p$ **and** $r_d > \epsilon_d$ **do**
      **while** $(\nabla_Y \mathcal{L}_\rho)^T (\nabla_Y^2 \mathcal{L}_\rho)^{-1}(\nabla_Y \mathcal{L}_\rho) > \epsilon$ **do**
         $\nabla_Y \mathcal{L}_\rho \triangleq -\nabla_Y \sum_t \log p(s_t|y_t) + \rho \mathcal{A}(Y) - \mathcal{A}^T(\rho Z - \Lambda)$
         $\nabla_Y^2 \mathcal{L}_\rho \triangleq -\nabla_Y^2 \sum_t \log p(s_t|y_t) + \rho I_{nT} - \rho \frac{1}{T}(\mathbf{1}_T \otimes I_n)(\mathbf{1}_T \otimes I_n)^T$
         $Y \leftarrow Y - (\nabla_Y^2 \mathcal{L}_\rho)^{-1}\nabla_Y \mathcal{L}_\rho$
      **end while**
      $U\Sigma V^T \triangleq \mathrm{SVD}(\mathcal{A}(Y) + \Lambda/\rho)$
      $Z' \leftarrow U \mathcal{S}_{\lambda\sqrt{nT}/\rho}(\Sigma)V^T$
      $\Lambda \leftarrow \Lambda + \rho(\mathcal{A}(Y) - Z')$
      $r_p \leftarrow ||\mathcal{A}(Y) - Z'||_F$
      $r_d \leftarrow \rho||\mathcal{A}^T(Z - Z')||_F$
      $\epsilon_p \leftarrow \sqrt{nT}\epsilon_{abs} + \epsilon_{rel} \max(||\mathcal{A}(Y)||_F, ||Z'||_F)$
      $\epsilon_d \leftarrow \sqrt{nT}\epsilon_{abs} + \epsilon_{rel}||\mathcal{A}^T(\Lambda)||_F$
      $Z \leftarrow Z'$
   **end while**
   **return** Y

---

## A.2    Algorithms for Stable Principal Component Pursuit

---

**Algorithm 2** Alternating Direction Method of Multipliers for Nuclear Norm minimization with connectivity (Stable Principal Component Pursuit) (Eq. 12)

---

   **input** Matrix of spike counts $S$ and spike histories $H$, learning rate $\rho$, parameters $\lambda$, $\gamma$
   $Y \leftarrow \log(S+1), Z \leftarrow 0, D \leftarrow 0, \Lambda \leftarrow 0$
   **while** $r_p > \epsilon_p$ **and** $r_d > \epsilon_d$ **do**
      **while** $(\nabla_Y \mathcal{L}_\rho)^T (\nabla_Y^2 \mathcal{L}_\rho)^{-1}(\nabla_Y \mathcal{L}_\rho) > \epsilon$ **do**
         $\nabla_Y \mathcal{L}_\rho \triangleq -\nabla_Y \sum_t \log p(s_t|y_t) + \rho \mathcal{A}(Y) - \mathcal{A}^T(\rho Z + \rho \mathcal{A}(DH) - \Lambda)$
         $\nabla_Y^2 \mathcal{L}_\rho \triangleq -\nabla_Y^2 \sum_t \log p(s_t|y_t) + \rho I_{nT} - \rho \frac{1}{T}(\mathbf{1}_T \otimes I_n)(\mathbf{1}_T \otimes I_n)^T$
         $Y \leftarrow Y - (\nabla_Y^2 \mathcal{L}_\rho)^{-1}\nabla_Y \mathcal{L}_\rho$
      **end while**
      $D \leftarrow \arg\min_D \gamma \frac{T}{n}||D||_1 + \frac{\rho}{2}||\mathcal{A}(DH) + Z - \mathcal{A}(Y) - \Lambda/\rho||_F^2$ (See Alg. 3)
      $U\Sigma V^T \triangleq \mathrm{SVD}(\mathcal{A}(Y - DH) + \Lambda/\rho)$
      $Z' \leftarrow U \mathcal{S}_{\lambda\sqrt{nT}/\rho}(\Sigma)V^T$
      $\Lambda \leftarrow \Lambda + \rho(\mathcal{A}(Y) - Z')$
      $r_p \leftarrow ||\mathcal{A}(Y - DH) - Z'||_F$
      $r_d \leftarrow \rho||\mathcal{A}^T(Z - Z')||_F$
      $\epsilon_p \leftarrow \sqrt{nT}\epsilon_{abs} + \epsilon_{rel} \max(||\mathcal{A}(Y - DH)||_F, ||Z'||_F)$
      $\epsilon_d \leftarrow \sqrt{nT}\epsilon_{abs} + \epsilon_{rel}||\mathcal{A}^T(\Lambda)||_F$
      $Z \leftarrow Z'$
   **end while**
   **return** Y

---

---

**Algorithm 3** Alternating Direction Method of Multipliers for Updating $D$ (Eq. 15)

---

   **input** Variables from Alg. 2, learning rate $\alpha$
   $E \leftarrow D, \Gamma \leftarrow 0$
   **while** $r_p > \epsilon_p$ **and** $r_d > \epsilon_d$ **do**
      **for** $i = 1 \rightarrow n$ **do**
         $D^i \leftarrow (\mathcal{A}^T(\mathcal{A}(Y^i) - Z^i + \Lambda^i/\rho)H^T + \alpha E_i - \Gamma_i)(\mathcal{A}(H)\mathcal{A}(H)^T + \alpha I_{nk})^{-1}$
      **end for**
      $E' \leftarrow \mathcal{S}_{\gamma T/n\rho\alpha}(D + \Gamma/\alpha)$
      $\Gamma \leftarrow \Gamma + \alpha(D - E')$
      $r_p \leftarrow ||D - E'||_F$
      $r_d \leftarrow \alpha||E - E'||_F$
      $\epsilon_p \leftarrow \sqrt{n^2 k}\epsilon_{abs} + \epsilon_{rel} \max(||D||_F, ||E'||_F)$
      $\epsilon_d \leftarrow \sqrt{n^2 k}\epsilon_{abs} + \epsilon_{rel}||\Gamma||_F$
      $E \leftarrow E'$
   **end while**
   **return** D

---

# B   Fitting Linear Dynamical Systems

It is not immediately obvious that we should fit linear dynamical systems by the particular subspace method used in this paper, or that we should minimize the nuclear norm of $\mathcal{A}(Y)$ instead of $Y$. Here we show empirical results on model data with two methods for fitting linear dynamical systems, and minimizing $||\mathcal{A}(Y)||_*$ versus $||Y||_*$, for a total of 4 combinations. The first method for fitting linear dynamical systems is perhaps the easiest. Let $X = (x_1, \ldots, x_T)$ be the matrix of latent states, just as $Y = (y_1, \ldots, y_T)$ is the matrix of natural rates, and suppose $x_t$ is generated by a linear dynamical system:

$$
\begin{aligned}
x_{t+1} &= Ax_t + \epsilon_t \\
\mathbb{E}[\epsilon_t] &= 0
\end{aligned}
\tag{27}
$$

First we take the singular value decomposition of $\mathcal{A}(Y)$, so that $U\Sigma V^T = \mathcal{A}(Y)$. Since $\mathcal{A}(Y) = CX$, the left and right singular vectors should be equal to $C$ and $X$ up to some arbitrary rotation $M$:

$$
\begin{aligned}
U\sqrt{\Sigma} &= CM \\
\sqrt{\Sigma}V^T &= M^{-1}X
\end{aligned}
\tag{28}
$$

where $\sqrt{\Sigma}$ takes the element-wise square root of the diagonal matrix of singular values.

It is clear that $X_{2:T} = (x_2, \ldots, x_T) = AX_{1:T-1} + E = A(x_1, \ldots, x_{T-1}) + (\epsilon_1, \ldots, \epsilon_{T-1})$, that is each column of $X$ is a noisy linear mapping of the column to the left of it. That suggests we could estimate $A$ by doing regression between past and future columns of $X$, or by proxy, $\sqrt{\Sigma}V^T$:

$$
\hat{A} = \left(V^{2:T}\sqrt{\Sigma^T}\right)\left(V^{1:T-1}\sqrt{\Sigma^T}\right)^{\dagger}
\tag{29}
$$

here $X^{i:j}$ denotes the $i$th to $j$th *rows* of $X$, while $X_{i:j}$ denotes the $i$th to $j$th columns. We refer to this method of fitting linear dynamical systems as *past-future regression*. While this estimate of $A$ is off by a change of coordinate, the eigenvalues should on average be the same as the true $A$ if our estimator is unbiased. We find that this is not the case.

Alternately, we use a variant of the Multivariable Output Error State sPace (MOESP) method for fitting linear dynamical systems, a type of subspace identification [26]. Our implementation of MOESP works as follows: take the covariance between $Y$ one and two time steps into the past and one and two time steps into the future:

$$
\Gamma = \begin{pmatrix} Y_{3:T-1} \\ Y_{4:T} \end{pmatrix}\begin{pmatrix} Y_{1:T-3} \\ Y_{2:T-2} \end{pmatrix}^T
\tag{30}
$$

where the matrix on the left is known as the block-Hankel matrix of future outputs and the matrix on the right is the block-Hankel matrix of past outputs in the terminology of subspace identification. The number of block-rows can be greater than 2, but as long as the number of latent dimensions is less than the number of observed dimensions, only two are needed.

From Eqs. 1 and 27 we can expand out $y_t$ as $CA^k x_{t-k} + \sum_{\tau=1}^{k} CA^{k-\tau}\epsilon_{t-\tau}$ and plug this into the true past-future covariance to find:

$$\text{Cov}\left[\left(\begin{array}{c} y_t \\ y_{t+1} \end{array}\right), \left(\begin{array}{c} y_{t-2} \\ y_{t-1} \end{array}\right)\right] = \left(\begin{array}{c} C \\ CA \end{array}\right)\left(\begin{array}{cc} & C\text{Cov}[x_t]A^{2T} \\ CA\text{Cov}[x_t]A^{2T} + C\text{Cov}[\epsilon_t]A^T \end{array}\right)^T \quad (31)$$

of which $\Gamma/(T-2)$ is the maximum likelihood estimate. We can then say the left singular values of $\Gamma$ should asymptotically be equal to $\left(\begin{array}{c} C \\ CA \end{array}\right)$ up to a rotation, and we estimate $\hat{A}$ by doing least squares regression between the top $n$ rows and bottom $n$ rows of the left singular vectors of $\Gamma$. As with the past-future regression method described above, we find it useful to scale the left singular vectors by the square root of the singular values.

In the experiments on model data, we know the true dimensionality $m$, and truncated $\Sigma$ so that all singular values after the $m$th are set to 0. On real data we would use some heuristic, such as truncating everything below the geometric mean of the first and last singular value. If we had access to known input as well, we would also include a projection onto the orthogonal subspace of the input, and include past inputs in the right-hand matrix in Eq. 30.

Figure 5: A comparison of the eigenvalues of the transition matrix $A$ recovered by different methods of fitting linear dynamical systems to model data. True eigenvalues in black, estimates over different trials in red. 200 neurons, 10000 time bins, 8 latent dimensions, 10 trials. Top row: Results of estimating $A$ by past-future regression (Eq. 29). Bottom row: Results of estimating the transition matrix by the MOESP subspace identification method. Left column: Nuclear norm minimization directly on the matrix $Y$. Right column: Nuclear norm minimization on the mean-centered matrix $\mathcal{A}(Y)$. In both cases the mean of $Y$ was subtracted before estimating the transition matrix. Note that both mean-centering and subspace identification are necessary to arrive at unbiased estimates of the transition matrix.

# C    Optimizing Smoothing Parameters

The only free parameter in our minimization is $\lambda$, which controls the tradeoff between the data likelihood and nuclear norm penalty. As $\lambda \to \infty$, the natural rates are forced to a low-rank solution, and eventually to the mean of the data (after being passed through the inverse nonlinearity) if the mean-centering operator is included, or zero if not. At the other extreme, the nuclear norm penalty vanishes as $\lambda \to 0$ and the natural rates go to the maximum likelihood solution. We demonstrate the effect of varying the nuclear norm penalty on the spectra of the natural rates and the eigenvalues of the recovered transition matrix and show that the nuclear norm penalty leads to less biased estimates of the transitions. As $\lambda$ increases the quality of the estimates improves, until the rank of the natural rates is forced to below the actual number of latent dimensions. In a nutshell, the solution should be as low rank as possible, but no lower.

Figure 6: Effect of varying the smoothing parameter $\lambda$ on the recovered transition matrix eigenvalues. True values in black, recovered in red. The nuclear norm term helps reduce the variance of the estimates, but the results quickly degenerate when $\lambda$ is too large. Note that the results are robust across a wide range of values of $\lambda$, from roughly 0.001 to 0.03. Model data, 200 neurons, 1000 time bins, 8 latent dimensions, 5 trials.

Figure 7: A simultaneous comparison of the eigenvalues of $A$ and singular values of $Y$ for various values of $\lambda$ on the same data as Fig. 6. Note that with no smoothing, the top singular values of the recovered $Y$ closely match the true values, but the noise leads to biases in the recovered $A$. At the other extreme, the quality of the recovered $A$ degrades rapidly if the smoothing forces the rank of $Y$ to be smaller than the true rank.