[Reviews · NeurIPS 2013]

Submitted by Assigned_Reviewer_4

The authors present a method for reducing the dimensionality of neural activity data. They take advantage of recent advances in applied mathematics to solve sparse and low-rank matrix approximation problems using convex relaxation, as well as of Bregman divergences for exponential families. They apply the method to synthetic data.

I find the work mathematically sound and the presentation generally clear. However, the authors may want to devote more attention to introducing the problem. Indeed, in the beginning of Section 2, they postulate the existence of a low-dimensional subspace of neuronal activity. It would seem to me that the dynamics in this low-dimensional subspace, Eq.(1), should be mediated by neurons and their synaptic connections. However, Eq.(1) does not include the synaptic connectivity matrix D, which instead appears in Eq.2. Because the authors point out that the low-dimensional dynamics would be present even in the absence of inputs (u_t = 0) I don’t understand what physical substrate underlies the dynamics of low-dimensional activity.

Additionally, while I admire the authors for their effort to introduce fashionable applied mathematics techniques into neuroscience, it is difficult to evaluate the value of their method until it is applied to actual data and generates insight into the function of a biological system.

Minor on line 291:
minimizing D -> minimizing over D
Summary: Dimensionality reduction method using convex relaxation of rank minimization

Submitted by Assigned_Reviewer_5

This paper combines a variety of standard techniques to describe and learn a generative model for multiple spike trains. They posit that neural activity can be generated via a combination of low dimensional latent variables and direct interactions. They learn the mapping from the low dimensional latent space to the high dimensional firing rate space by imposing a nuclear norm penalty, and they learn the sparse interactions by imposing an L1 penalty, similar to stable principal component pursuit. To minimize the nuclear norm, they use ADMM and to learn the various parameters they do coordinate descent. They show that they can recover the parameters from data sampled from their own model.

Overall, while the combination of techniques is interesting, this paper feels like that is all there is - just a combination of existing techniques. They do not apply their algorithm to real data (a strong requirement in my opinion) and show that it yields scientific insight, above and beyond previous models. Overall, I do not believe this paper is a sufficient conceptual advance over previous work.
Summary: An interesting combination of existing generative models and fitting techniques, but not a sufficient conceptual advance, and no application to real data.

Submitted by Assigned_Reviewer_6

Summary:
The author(s) proposed a method for dimensionarity reduction method that allows multiple dynamical systems therein. The method is a natural extension of PCA and uses the alternating direction method to solve a convex but non-smooth problem. The experiments show the superiority of the method.

Quality:
The generative model is appropriate for modeling neural activities and the proposed method for estimating the parameters in the model is reasonable and novel.

Clarity:
The manuscript is well written and easy to understand.

Originality:
The work is original enough although it uses several existing methods.

Significance:
The proposed method has extensions and can be used in a wide range of applications.
Summary: This work is based on a good model and an appropriate estimation method. The proposed method worked well in the experiments.
Author Feedback

Author rebuttal: Reviewer 4:

>The authors may want to devote more attention to introducing the problem. Indeed, in the beginning of Section 2, they postulate the existence of a low-dimensional subspace of neuronal activity. It would seem to me that the dynamics in this low-dimensional subspace, Eq.(1), should be mediated by neurons and their synaptic connections. However, Eq.(1) does not include the synaptic connectivity matrix D, which instead appears in Eq.2. Because the authors point out that the low-dimensional dynamics would be present even in the absence of inputs (u_t = 0) I don’t understand what physical substrate underlies the dynamics of low-dimensional activity.

There are many theoretical and experimental results supporting the existence of low-dimensional dynamics in some neural systems. The citations provided in the introduction all use dimensionality reduction techniques to find low-dimensional network states. Some of these network states are correlated with behavioral or perceptual activity, others seem to reflect intrinsic dynamics, but in all cases they demonstrate the presence of interesting low-dimensional structure. There are also results from physiology showing highly correlated activity in large networks [e.g. Schneidman et al, Nature (2006)] suggesting low-dimensional models can account for most of the variability in these networks. On the theoretical side, there is a rich literature showing how low-dimensional attractors can emerge from the activity of large networks. We apologize if this was not explained clearly enough in the introductory section, and we can improve that in the final manuscript.

>Additionally, while I admire the authors for their effort to introduce fashionable applied mathematics techniques into neuroscience, it is difficult to evaluate the value of their method until it is applied to actual data and generates insight into the function of a biological system.

We agree that a demonstration on real neural data would significantly strengthen the case made in our paper. Since submission, we have tried our approach on several real neural datasets with promising results. We are most excited about results on data from the group of Nicho Hatsopoulos. The data consists of spike times from 125 neurons in macaque motor cortex during a two dimensional reaching task. We binned spikes every 100 ms and estimated linear dynamical systems by the same procedure used to generate the results in Fig. 2., using 8000 time bins of training data. We evaluated the approximate log likelihood on 2000 time bins of held out data using Laplace filtering. Nuclear norm smoothing significantly improved the held out log likelihood of the best linear model (~750 bits/s) over fitting without strong nuclear norm penalty (~1200 bits/s), and the best linear model required fewer dimensions (~15) than the best linear model without strong nuclear norm penalty (~35). We plan to include these results in the revised paper.

>Minor on line 291: minimizing D -> minimizing over D

We thank the reviewer and will correct that in the final manuscript.

Reviewer 5:

>To learn the various parameters they do coordinate descent.

The parameters of the dynamical system were actually learned using spectral methods applied to the estimate of low-dimensional latent activity. We can rewrite this part of the paper to make that more clear if necessary.

>They do not apply their algorithm to real data (a strong requirement in my opinion) and show that it yields scientific insight, above and beyond previous models.

Please see our response to reviewer 4 on the same point.

We'd also like to bring attention to two sections of the paper that the reviewers declined to comment on, and therefore we believe were not highlighted enough in the original manuscript. First, evaluating the performance of exponential family PCA by use of Bregman divergence explained, analogous to variance explained in regular PCA, is not something that we have seen elsewhere in the dimensionality reduction literature. While the mathematics behind it is not new, we believe that it is a simple and useful method for evaluating low-dimensional exponential family models that deserves wider attention.

Second, we believe our sparse+low rank approach is the first convex method to address the common input problem in connectivity inference. Numerous studies have used sparse or group-sparse penalties to smooth networks fit to spiking activity, both on real data [Pillow et al, Nature (2008)], and model data [Stevenson et al, IEEE Trans. Neural Systems and Rehabilitation (2009)], [Hertz, Roudi and Tyrcha, arXiv (2011)]. Recent work [Gerhard et al, PLoS Comp Bio (2013)] has validated these methods on real biological neural networks with known connectivity. These networks are small and fully-observed, so no regularization is needed. We show that in some cases a sparsity penalty alone gives no benefit in the presence of common inputs, while a combination of low rank and sparsity penalty does, so long as the common inputs are relatively low dimensional. This is a novel result in the modeling literature and one that is likely to have important practical benefit as connectivity inference methods are scaled to more complex neural systems. While we would love to validate our approach to connectivity inference on real data, there are not yet data at the relevant scale where ground truth connectivity is known.

We appreciate that this point may not have come across clearly enough in the manuscript as it's now organized. While the first section of the paper focused on learning models of the dynamics of common inputs, the focus of the experimental section on sparse+low rank penalties was on removing the confounding effect of common inputs, and some context may have been lost in the transition. We will rewrite the paper to better contextualize and highlight the importance of these results.